# Multimorbidity Patterns among People with Type 2 Diabetes Mellitus: Findings from Lima, Peru

**DOI:** 10.3390/ijerph19159333

**Published:** 2022-07-30

**Authors:** Antonio Bernabe-Ortiz, Diego B. Borjas-Cavero, Jimmy D. Páucar-Alfaro, Rodrigo M. Carrillo-Larco

**Affiliations:** 1CRONICAS Center of Excellence in Chronic Diseases, Universidad Peruana Cayetano Heredia, Lima 15074, Peru; r.carrillo-larco@imperial.ac.uk; 2School of Medicine, Universidad Científica del Sur, Lima 15067, Peru; 3Hospital de Emergencias Villa El Salvador, Lima 15837, Peru; diego.borjas@upch.pe (D.B.B.-C.); jimmydapmd@gmail.com (J.D.P.-A.); 4Department of Epidemiology and Biostatistics, School of Public Health, Imperial College London, London SW7 2AZ, UK; 5Universidad Continental, Lima 15046, Peru

**Keywords:** Type 2 diabetes mellitus, phenotypes, electronic health records, k-means

## Abstract

Background: Type 2 diabetes (T2DM) is a chronic condition with a high disease burden worldwide, and individuals with T2DM often have other morbidities. Understanding the local multimorbidity profile of patients with T2DM will inform precision medicine and public health, so that tailored interventions can be offered according to the different profiles. Methods: An analysis was conducted of electronic health records (2016–2021) in one hospital in Lima, Peru. Based on ICD-10 codes and the available measurements (e.g., body mass index), we identified all T2DM cases and quantified the frequency of the most common comorbidities (those in ≥1% of the sample). We also conducted k-means analysis that was informed by the most frequent comorbidities, to identify clusters of patients with T2DM and other chronic conditions. Results: There were 9582 individual records with T2DM (mean age 58.6 years, 61.5% women). The most frequent chronic conditions were obesity (29.4%), hypertension (18.8%), dyslipidemia (11.3%), hypothyroidism (6.4%), and arthropathy (3.6%); and 51.6% had multimorbidity: 32.8% had only one, 14.1% had two, and 4.7% had three or more extra chronic conditions in addition to T2DM. The cluster analysis revealed four unique groups: T2DM with no other chronic disease, T2DM with obesity only, T2DM with hypertension but without obesity, and T2DM with all other chronic conditions. Conclusions: More than one in two people with T2DM had multimorbidity. Obesity, hypertension, and dyslipidemia were the most common chronic conditions that were associated with T2DM. Four clusters of chronic morbidities were found, signaling mutually exclusive profiles of patients with T2DM according to their multimorbidity profile.

## 1. Introduction

Type 2 diabetes mellitus (T2DM) is one of the most important public health issues worldwide. The number of people living with this condition has quadrupled between 1980 and 2014 [1], and will reach 643 million by 2030 [2]. Globally, the total health expenditure due to T2DM has been estimated to be USD 966 billion dollars by the International Diabetes Federation [2]. In Peru, a middle-income country that is located in South America, the national prevalence of T2DM has increased from 5% in 2005 [3] to 7% in 2012 [4,5], though these country-level prevalence hide a great heterogeneity within the country, with regions having estimates over 10% [5,6].

T2DM is a chronic condition which is often found with other chronic diseases even among younger subjects [7,8]. The presence of two or more chronic conditions in the same individual is known as multimorbidity, and the current evidence shows that multimorbidity is more common among people with T2DM, amongst whom the prevalence of multimorbidity may range between 30% to almost 90% [9,10,11]. In addition, multimorbidity increases the number of hospital visits (outpatient and inpatient), augments the use of specialized health resources, leads to polypharmacy (i.e., the increase in the number of taken medications), with the subsequent increase of health-related costs [12,13] and burden of treatment [14].

A previous work using information from the Basque Health Service in Spain found that among 125,000 subjects with T2DM, some chronic conditions tend to cluster, reporting a total of 10 potential groups of morbidities [9]. Among these clusters, cardiovascular conditions (hypertension, ischemic heart disease, atrial fibrillation, chronic heart disease, heart failure, and chronic kidney disease) were the most relevant. However, a work using the Clinical Practice Research Datalink from the United Kingdom and more than 100,000 T2DM cases reported only two clusters of morbidities, and they were consistent when the sample was split by years of T2DM diagnosis [15]. These results suggest that clusters of morbidities depend on several factors, including, but not limited to, health system infrastructure and capacity and the national or subnational context profile of other chronic diseases (i.e., tuberculosis may be relevant in resource-constrained settings, but not in high income countries), among others.

Electronic health records (EHR) are a unique opportunity to evaluate the most important combinations of chronic morbidities [16], especially among those with non-communicable diseases such as T2DM. Information that is derived from EHR may provide a better understanding of the nature, prevalence, and patterns of morbidities among T2DM patients. This may be key to guarantee an appropriate management and follow-up of these cases, particularly relevant in resource-constrained settings. As a result, in this study, we explored the patterns of morbidities and the prevalence of multimorbidity among cases with T2DM diagnosis using information from a second-level health facility in the South of Lima, Peru. In addition, we tried to identify the most frequent clusters of such chronic conditions.

## 2. Methods

### 2.1. Data Sources

We analyzed the EHR of patients available in the Hospital de Emergencias Villa El Salvador (HEVES), in Lima, Peru. This hospital is a second-level healthcare facility with a target population of almost 1 million people from eight districts in the Southern Lima [17]. It is currently a reference hospital providing specialized care and medical and surgical attention of intermediate complexity.

The electronic records (as inpatient or outpatient) were extracted since the system inception in 2016 to December 2021. We included only Peruvian subjects with a diagnosis of T2DM according to the International Classification of Diseases 10th revision [ICD-10], code: E11 [18] in at least one of the health visits. The subjects that were included were those aged ≥18 years, with information that was recorded in the EHR system. In addition, cases with a E10, E12, and E13 code were also excluded to discard double diabetes diagnoses. Information from the patients (including sociodemographic and other morbidities diagnosis) and the laboratory results were merged and analyzed.

### 2.2. Definition of Variables

Morbidities that were included in the analyses were anemia, anxiety, arthropathy, atrial fibrillation, cancer, cerebrovascular disease, chronic back pain, chronic kidney disease, chronic obstructive pulmonary disease (COPD), dementia (including Alzheimer), depression, dyslipidemia, ischemic heart disease, hypertension, hypothyroidism, heart failure, obesity, tuberculosis, and urinary lithiasis. The list of morbidities that were selected was based on the most common chronic conditions. We excluded T2DM complications from the list of morbidities as they were underreported. The definitions were based on ICD-10 codes as well as anthropometry measurements and laboratory results (i.e., obesity, hypertension, and dyslipidemia). For ICD-10 codes, the diagnosis of morbidities was built using the information of all the health visits that were available. In the case of anthropometry and laboratory information, as these may have been measured in different health visits, diagnosis was based on information from all the visits. For example, in the case of obesity, the ICD-10 code (E66) was used if available, but also the body mass index (BMI) by utilizing the height and weight that were measured during different health visits. Details for other conditions are shown in Appendix A.

Multimorbidity was defined by adding up the most common chronic conditions (i.e., those with a prevalence ≥ 1%), and then split into groups (1, 2 or 3+ morbidities). Thus, subjects with a diagnosis of T2DM with an extra morbidity (i.e., ≥1 chronic condition) had multimorbidity. Sociodemographic variables that were included in the analysis were sex (women vs. men) and age (<40, 40–49, 50–59, 60–69, and 70+ years).

### 2.3. Statistical Analysis

The analyses were conducted using STATA v16 for Windows (Stata Corp, College Station, TX, USA) and *p* < 0.05 was considered statistically significant. Initially, we described the subjects’ characteristics using the mean and standard deviation (SD) or median and interquartile range (IQR) for numerical variables, according to normal distribution that was evaluated using Shapiro–Wilk test; we utilized frequencies and proportions for categorical variables. A detailed description of the morbidities for the overall sample and by sex was tabulated, and a Chi-squared test was used for a comparison between men and women.

Using the most common chronic morbidities (i.e., those ≥ 1% in the overall sample), a Nightingale Rose chart was utilized to compare the distribution of chronic morbidities considering the total number of cases and their respective proportions. In addition, clusters were created using a partition-clustering approach (k-means) for observations. Under this approach, the observations are assigned to the group with the closest center. The mean of the observations that were assigned to each of the groups is computed, and through an iterative process, these steps continue until all the observations remain in the same group from the previous iteration [19]. As binary outcomes were evaluated (i.e., conditions with a no/yes option), the Jaccard binary similarity coefficient was utilized [20]. The number of clusters were defined by using the Calinski/Harabasz pseudo-F test for non-hierarchical cluster analysis [21], but also based on the clinical relevance comparing from two to six potential clusters. Finally, the clusters were characterized according to age, sex, and the distribution of morbidities.

### 2.4. Ethics

The study protocol was approved by the Ethical Committee of the Hospital de Emergencias Villa El Salvador (HEVES), in Lima, Peru (Exp 001-2022, registration code: 22-001475-001). For analysis purposes, electronic clinical records were deidentified to guarantee confidentiality and anonymity of patients.

## 3. Results

### 3.1. Characteristics of the Study Population

From 2016 to 2021, 9582 individual records with diagnosis of T2DM were available and included in the analyses, with a mean age of 58.6 (SD: 13.5) years, 5896 (61.5%) females, and a median of 6 visits over time (IQR: 2–12). The most frequent chronic conditions were obesity (29.4%), hypertension (18.8%), dyslipidemia (11.3%), hypothyroidism (6.4%), arthropathy (3.6%), chronic kidney disease (2.0%), anemia (1.8%), chronic back pain (1.7%), and anxiety (1.2%). According to sex, obesity, dyslipidemia, hypothyroidism, arthropathy, anxiety, and depression were the most common chronic conditions among women. On the other hand, kidney chronic disease, cerebrovascular disease, tuberculosis, and heart ischemic disease were most common among men (Table 1).

### 3.2. Multiple Chronic Conditions

The number of morbidities among patients with T2DM had a median of 1 (range: 0–7). A total of 4945 (51.6%) subjects with T2DM had multimorbidity (i.e., at least one extra chronic condition); 3145 (32.8%) presented one extra condition only (i.e., T2DM and one more condition); whereas 1351 (14.1%) had two conditions; and 449 (4.7%) presented three or more chronic conditions in addition to T2DM.

Multimorbidity was more frequent among women (56.9%) compared to men (43.1%, *p* < 0.001). Similarly, multimorbidity was more frequent among older subjects (54.6% among subjects aged 70+ years) compared to younger ones (47.9% among those < 40 years, *p* < 0.001). Details of the distribution of morbidities by the number of chronic conditions are shown in Figure 1 and Appendix A.

### 3.3. Cluster Analysis

Based on the Calinski/Harabasz test and clinical relevance, four groups were found after cluster analysis (Table 2). One group comprised of subjects with T2DM and no other extra conditions (Cluster 2). Another group comprised all the cases of T2DM with obesity (Cluster 3). A third group comprised of subjects with cardiovascular conditions, especially hypertension, but without obesity (Cluster 4). Finally, the fourth group included all other chronic conditions without including obesity and hypertension (Cluster 1). Cluster 2 was characterized by a greater proportion of men compared to the other clusters, whilst Cluster 1 had the highest proportion of women. On the other hand, Cluster 4 was comprised of older individuals (60+ years) whereas Cluster 3 had the younger subjects (<50 years).

## 4. Discussion

### 4.1. Main Findings

Our results revealed that obesity, hypertension, and dyslipidemia were the more common chronic conditions among people with T2DM diagnosis. Using the most common chronic conditions, more than 50% of subjects with T2DM had multimorbidity, being more frequent among women than men, and among elderly. Finally, four clusters of chronic morbidities were found: T2DM alone (i.e., without any other condition), T2DM with obesity, T2DM with cardiovascular profile but without obesity (i.e., with hypertension), and T2DM with any other chronic conditions.

### 4.2. Comparison with Previous Studies

Evidence shows a great variation in the prevalence of multimorbidity among people with T2DM, highlighting that much of the previous research has been conducted in high-income countries. Thus, according to a systematic review, data-driven algorithms that were used in different manuscripts point out to a large heterogeneity in T2DM subtypes with uncertainty on the variables that should be utilized for creating phenotypes of T2DM cases [22]. For example, a study with T2DM cases aged ≥35 years, with at least one year of diagnosis, and from a large primary healthcare cohort in the UK, found an age-standardized prevalence of multimorbidity of 33%; and hypertension was the most common morbidity in the overall sample, whilst depression was the second most prevalent condition among women, and coronary heart disease among men [15]. In addition, this manuscript also demonstrated that morbidity patterns that were associated with T2DM had important changes during follow-up. Thus, mental health problems, mainly depression prevalence, increased over time. Pouplier et al., using longitudinal data of subjects with T2DM, aged ≥40 years, that were newly diagnosed during the previous 2 years, and from a cluster randomized trial that was conducted in Denmark, reported that 32% of people with T2DM had a multimorbidity at the moment of diabetes diagnosis, and this proportion increased to 80% after 16 years of follow-up [23]. In this latter study, the most prevalent chronic conditions were cardiovascular and musculoskeletal diseases among surviving cases, and multimorbidity was more common among older individuals and those living alone. However, a study using information of 424 T2DM subjects that were diagnosed for at least one year, aged 18 years and over, from a randomized controlled trial that was conducted in Ireland, reported that 90% had at least one additional chronic condition; 74% had a circulatory condition, whereas 36% had a metabolic/endocrine/nutritional condition in addition to T2DM [24]. In addition, in this study, the number of clinical visits as well as polypharmacy increased with the number of chronic conditions. Similarly, in Spain, using data of adults ≥35 years from primary healthcare (i.e., Basque Health Service), 90% of subjects with a diagnosis of T2DM had at least one other chronic condition, and cardiovascular diseases (i.e., hypertension, ischemic heart disease, atrial fibrillation, renal failure, and heart failure) was the most common cluster, whereas the second one included mental health problems (anxiety, depression, and somatoform disorders) [9]. However, in Spain as well, when younger subjects from a population-based study in Catalonia were included (≥18 years) in the sample, the prevalence of multimorbidity among T2DM patients diluted to 53%, but similar to previous reports, complicated hypertension and atherosclerosis/peripheral vascular disease were the most frequent conditions in these patients, followed by cholecystitis and cholelithiasis, retinopathy, and peripheral neuropathy [25].

Some studies have been conducted in resource-constrained settings. From a list of 27 chronic conditions, hypertension, coronary atherosclerosis and other heart diseases, as well as acute cerebrovascular disease were the top three comorbidities among adult patients that were hospitalized (i.e., based on hospital discharge forms) with a diagnosis of T2DM in Northeast China [26]. Gao et al., using latent class analysis with records from 10 hospitals, reported four clusters using information of complications and comorbidities among people with T2DM. Across these groups, differences in demographics, diabetes severity, and behavioral factors were found [27]. Nevertheless, women had an increased prevalence of hypertension, dyslipidemia, and cardiovascular disease compared with men. Finally, the prevalence of multimorbidity was 74%, with hypertension (59%), dyslipidemia (37%), arthritis (20%), cardiovascular disease (9%), and thyroid disease (9%) being the most frequent conditions among 400 people with T2DM from a community-based cross-sectional survey in Kerala, India [11].

Overall, our results are in line with different studies from low-, middle-, and high-income countries. Based on the aforementioned research manuscripts, our findings highlight an elevated prevalence of multimorbidity among people with T2DM diagnosis. In addition, hypertension was the leading comorbid condition that was associated with T2DM.

### 4.3. Relevance for Public Health

Multimorbidity is a relevant topic given its high prevalence among people with T2DM. Some clusters of morbidities are more prevalent in some subpopulations. For example, the cluster of T2DM with obesity may imply recent diabetes diagnosis as this was the youngest group, or a poor treatment adherence. In addition, obesity has not been included in analysis by previous reports, and this condition may highlight the concomitant need of an aggressive approach and management to avoid future multimorbidity and T2DM-related complications. On the other hand, T2DM patients with a high risk of cardiovascular conditions (i.e., those having hypertension but not obesity) may require a more intensive treatment above and beyond glycemic control; that is, cardio-renal protection should be warranted for these patients. T2DM is accompanied by the occurrence of other chronic morbidities, which may have an impact on disease management, with the subsequent increase in the number of medications that are needed, clinic visits, and health costs [13]. In that line, a systematic review reported that multimorbidity among those with T2DM has been associated with all-cause mortality and hypoglycemia events despite of the heterogeneity in the studies’ sample size as well as the methodology that was used to assess multimorbidity [28]. Hypoglycemia in the case of multimorbidity may reveal problems of over-treatment or very aggressive therapy for those with multiple chronic conditions. Thus, the burden of treatment can be high among those with T2DM and multimorbidity, and unexpected complications may arise.

The use of cluster analysis allows a better characterization of multimorbidity patterns by identifying subsets of the T2DM population that may need different demands from the health system [7], but also it can provide an understanding of potential etiologies that are related to T2DM morbidity. In this line, T2DM cases with obesity may imply recent diagnosis which could require a different management (i.e., number and type of medications, number of control visits, etc.) than those having an increased cardiovascular risk (T2DM with hypertension). Although Peru’s healthcare system has achieved progress towards “health for all” (universal health coverage) [29], healthcare needs, especially for those with T2DM, particularly in regard to meeting the needs of those with multimorbidity, are inadequate due to the high fragmentation and segmentation of services to provide a full continuum of care as well as the lack of specific national/subnational programs to support T2DM patients. Moreover, the local healthcare workforce has major capacity limitations in relation to quantity, quality, and equitable distribution, which may be more common in rural and semiurban areas. Finally, there is a poor-inter-operability of health information systems [30]. Thus, multimorbidity in people with T2DM will challenge the performance of the healthcare system in Peru if appropriate strategies are not taken.

The current Clinical Guideline for the Diagnosis, Treatment, and Control for Type 2 Diabetes Mellitus in the Primary Healthcare in Peru, published in 2016 [31], establishes the process and number of clinical evaluations a patient with T2DM should follow after diagnosis. However, much of these assessments and tests are not part of the primary care and requires the visit of a second- or third-level health facility, which may complicate access to appropriate management and adherence to different treatments. An adequate strategy should be implemented to guarantee the diagnosis as well as the management of different chronic conditions (i.e., multimorbidity) among subjects with T2DM.

### 4.4. Strengths and Limitations

This study benefitted from the use of electronic health records and several years of information from a reference hospital in Lima, Peru. Nevertheless, some limitations highlight the need of discussion. First, selection bias may be present as EHR from a single hospital was used. Moreover, the HEVES is a second-level healthcare facility, and thus, more severe cases (i.e., those with complex morbidities) may not be fully captured as they would need more specialized attention (i.e., third-level care attention), with a subsequent underestimation of the prevalence of multimorbidity. Despite of that, our results are in line with aforementioned reports. Second, detection bias can also be an issue. As T2DM patients with longer time of disease and more chronic conditions may have more regular health visits and, for instance, have more morbidities that are detected compared to those with a recent diagnosis or chronic conditions different from T2DM. Additionally, the quality of data, although relevant for analysis, can be an issue as some common conditions (i.e., chronic back pain, depression, or anxiety) appears very low. Third, only more common chronic conditions were considered from our list of morbidities. Thus, we included tuberculosis but decided to exclude other infectious diseases because of their low frequency in our study population. We also excluded T2DM complications from the list of morbidities as a great underreporting was found (i.e., complications are not recorded as ICD-10 code). In addition, unlike other studies, we included obesity in the list of chronic morbidities, which we believe allows a better characterization of people with T2DM which has a strong obesogenic component, especially in a country that is undergoing the nutrition transition. Similarly, only chronic conditions with a prevalence ≥ 1% were used for defining multimorbidity and clustering analysis. When all the lists of chronic conditions were used to estimate the prevalence of multimorbidity, this estimate reached up to 53.7% (data not shown). Despite that, the clusters in our study were in line with previous reports. Fourth, the COVID-19 pandemic may also have had an impact on our estimates because patients with different chronic conditions, but especially T2DM, did not have access to the hospital as often as they used to because the health system saturation with COVID-19 patients (relevant for years 2020 and 2021) [32]. Finally, cluster analysis is an exploratory strategy approach and, as a result, our findings may depend on the algorithm that was used. Nevertheless, the methodology that we used has become a common practice in the scientific literature about T2DM, and the findings that are reported are consistent with existing evidence.

## 5. Conclusions

More than half of people with T2DM had multimorbidity, and obesity, hypertension, as well as dyslipidemia, were the most common chronic conditions that were present among subjects with T2DM. A total of four clusters of chronic morbidities were found: T2DM alone, T2DM with obesity, T2DM with cardiovascular profile but without obesity (i.e., with hypertension), and T2DM with any other chronic conditions, signaling a mutually exclusive profile of patients with T2DM according to their multimorbidity profile, and suggesting the need of different management strategies for each of these profiles.

## Figures and Tables

**Figure 1 ijerph-19-09333-f001:**
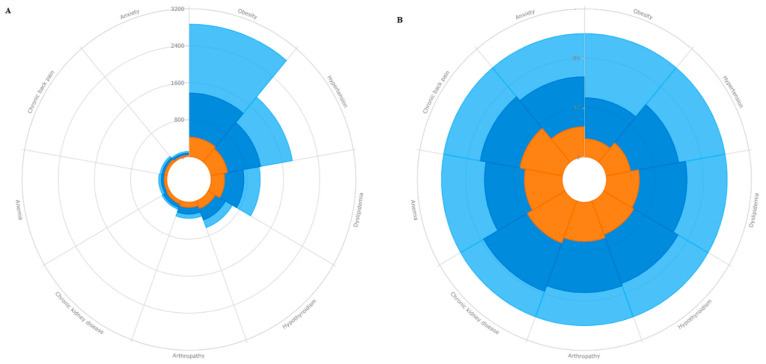
Distribution of morbidities by number of chronic conditions among T2DM cases: (**A**) The absolute number of cases adding up to the total sample size and (**B**) the relative frequencies related to all subjects within the respective comorbidity. Colors imply the number of chronic conditions in addition to T2DM: light blue (1 chronic condition), dark blue (2 chronic conditions), and orange (3+ chronic conditions).

**Table 1 ijerph-19-09333-t001:** Morbidities distribution among T2DM individuals: overall and by sex.

	Overall Sample	By Sex		*p*-Value
		Females	Males	
	(*n* = 9582)	(*n* = 5896)	(*n* = 3686)	
**Age**				0.01
<40 years	877 (9.2%)	551 (9.4%)	326 (8.8%)	
40–49 years	1577 (16.5%)	940 (15.9%)	637 (17.3%)	
50–59 years	2539 (26.5%)	1604 (27.2%)	935 (25.4%)	
60–69 years	2608 (27.2%)	1632 (27.7%)	976 (26.5%)	
70+ years	1981 (20.6%)	1169 (19.8%)	812 (22.0%)	
**Morbidity**				
Obesity	2823 (29.5%)	1979 (33.6%)	844 (22.9%)	<0.001
Hypertension	1796 (18.7%)	1117 (19.0%)	679 (18.4%)	0.52
Dyslipidemia	1084 (11.3%)	786 (13.3%)	298 (8.1%)	<0.001
Hypothyroidism	616 (6.4%)	547 (9.3%)	69 (1.9%)	<0.001
Arthropathy	345 (3.6%)	275 (4.7%)	70 (1.9%)	<0.001
Chronic kidney disease	191 (2.0%)	104 (1.8%)	87 (2.4%)	0.04
Anemia	172 (1.8%)	117 (2.0%)	55 (1.5%)	0.08
Chronic back pain	160 (1.7%)	107 (1.8%)	53 (1.4%)	0.16
Anxiety	111 (1.2%)	79 (1.3%)	32 (0.9%)	0.04
Cerebrovascular disease	79 (0.8%)	35 (0.6%)	44 (1.2%)	0.002
Tuberculosis	73 (0.8%)	27 (0.5%)	46 (1.3%)	<0.001
Cancer	69 (0.7%)	46 (0.8%)	23 (0.6%)	0.38
Heart ischemic disease	60 (0.6%)	24 (0.4%)	36 (1.0%)	<0.001
Atrial fibrillation	52 (0.5%)	25 (0.4%)	27 (0.7%)	0.05
Heart failure	49 (0.5%)	28 (0.5%)	21 (0.6%)	0.53
Urinary lithiasis	40 (0.4%)	28 (0.5%)	13 (0.3%)	0.27
Depression	16 (0.2%)	14 (0.2%)	2 (0.1%)	0.03
COPD	9 (0.1%)	4 (0.1%)	5 (0.1%)	0.29
Dementia	4 (<0.1%)	1 (<0.1%)	3 (0.1%)	0.13

COPD = Chronic obstructive pulmonary disease.

**Table 2 ijerph-19-09333-t002:** Cluster analysis (k-means): Characterization of clusters.

	Clusters			
	1	2	3	4
	(*n* = 952)	(*n* = 4637)	(*n* = 2823)	(*n* = 1170)
**Sex**				
Women	694 (72.9%)	2539 (54.8%)	1979 (70.1%)	684 (58.5%)
Men	258 (27.1%)	2098 (45.2%)	844 (29.9%)	486 (41.5%)
**Age**				
<40 years	91 (9.6%)	457 (9.9%)	315 (11.2%)	14 (1.2%)
40–49 years	156 (16.4%)	785 (16.9%)	568 (20.1%)	68 (5.8%)
50–59 years	257 (27.0%)	1288 (27.8%)	773 (27.4%)	221 (18.9%)
60–69 years	261 (27.4%)	1208 (26.1%)	746 (26.4%)	393 (33.6%)
70+ years	187 (19.6%)	899 (19.3%)	421 (14.9%)	474 (40.5%)
**Morbidities**				
Obesity	0 (0.0%)	0 (0.0%)	2823 (100.0%)	0 (0.0%)
Hypertension	0 (0.0%)	0 (0.0%)	677 (24.0%)	1119 (95.6%)
Dyslipidemia	477 (50.1%)	0 (0.0%)	483 (17.1%)	124 (10.6%)
Hypothyroidism	274 (28.8%)	0 (0.0%)	290 (10.3%)	52 (4.4%)
Arthropathy	130 (13.7%)	0 (0.0%)	158 (5.6%)	57 (4.9%)
Chronic kidney disease	3 (0.3%)	0 (0.0%)	50 (1.8%)	138 (11.8%)
Anemia	88 (9.2%)	0 (0.0%)	39 (1.4%)	45 (3.9%)
Chronic back pain	72 (7.6%)	0 (0.0%)	62 (2.2%)	26 (2.2%)
Anxiety	61 (6.4%)	0 (0.0%)	34 (1.2%)	16 (1.4%)
**Sum of morbidities**				
None	0 (0.0%)	4637 (100.0%)	0 (0.0%)	0 (0.0%)
1	809 (85.0%)	0 (0.0%)	1506 (53.4%)	830 (70.9%)
2	133 (14.0%)	0 (0.0%)	933 (33.1%)	285 (24.4%)
3+	10 (1.0%)	0 (0.0%)	384 (13.6%)	55 (4.7%)

## Data Availability

Restrictions apply to the availability of these data. The data were obtained from Hospital de Emergencias Villa El Salvador (HEVES) and are available with the permission of staff of the Hospital de Emergencias Villa El Salvador (HEVES).

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
