# Peer review of "Multimorbidity Patterns among People with Type 2 Diabetes Mellitus: Findings from Lima, Peru"

_ijerph, 2022, doi:10.3390/ijerph19159333_

Round 1
Reviewer 1 Report
This manuscript examines prevalence estimates and most frequent clusters of comorbidities among people with type 2 diabetes from Lima, Peru. Strengths mainly include the inclusion of electronic health records over several years (2016-2021) and the large age-range (people aged at least 18 years). The manuscript is well-structured and well-written. However, some questions remain:
Major comments:
1) Type 2 diabetes definition:
a. According to the methods section, only subjects with type 2 diabetes diagnosis were included in the analysis. The listed underlying ICD-10 codes are, however, E11, E12, E13, and E14. In fact, only E11 documents a type 2 diabetes diagnosis, whereas the other codes represent other specific diabetes forms. Therefore, at least a rationale and discussion regarding the inclusion of E12-14 is necessary and a sensitivity analysis with exclusion of persons with E12-E14 codes alone would be preferable.
b. Please explain how double diabetes diagnoses for the same person, which relate to different types of diabetes, i.e. particularly E11 (type 2 diagnosis) and E10 (type 1 diabetes), were handled for the present analysis.
c. Are there any information available in the database enabling the differentiation of duration and/or severity of type 2 diabetes, e.g. based on ICD-10 codes in consecutive years, laboratory values (HbA1c, glucose) or medications (none, oral antidiabetics, insulin), which can be used to further describe the identified clusters?
2) Selection of included comorbidities:
a. Please provide a hypothesis/rationale for initial selection of the comorbidities.
b. Specifically, some diabetic-specific complications are relatively frequent (at least not <1%, which was the cut-off for further consideration for the analysis), such as diabetic polyneuropathy, retinopathy and nephropathy. While nephropathy was considered by including chronic kidney disease (N18), the other complications are not addressed at all. Is it possible to also include the further complications in the analysis? Otherwise, please discuss their exclusion as limitation.
3) Table 1: Was the age-distribution similar among females and males in the sample? Preferably, please include age (i.e., the age-categories as displayed in table 2) as a descriptive variable in Table 1.
4) Please discuss the quality/validity of used data from electronic health records. For example, the prevalence of chronic back pain (1.7% according to Table 2) appears very low.
5) Discussion section “4.2 Comparison with previous studies”:
a. For a better interpretation of the mentioned comorbidity prevalences in other studies, please add the respective target population (e.g. population-/ community-based sample, hospitalised patients), the age-range and the disease severity (newly diagnosed cases of type 2 diabetes or cases from a long-term follow up) of included participants for each cited study, as these are related factors of comorbidities and could explain at least partially the reported variation in comorbidity prevalences.
b. In the last sentence “an increased prevalence of multimorbidity among people with T2DM diagnosis” is highlighted. What is the comparison group for this conclusion? If people without T2DM are meant, how is the prevalence of the same morbidities among the people without vs. with T2DM - in this study and in other studies?
Minor comments:
6) Please provide more information on measurements (i.e., anthropometry, blood pressure, and laboratory data) in the methods section.
7) At the end of the last paragraph in the introduction there is a duplication (“we explored the patterns of (multi)morbidities”), please correct.
8) For a better understanding of the Figure 1, please specify the legend description by naming numerator and denominator of A) and B). If I understand that right, in A) absolute numbers adding up to total sample size and in B) relative frequencies related to all persons within the respective comorbidity are shown.
9) Related to cluster 2 it is described that it “comprised subjects with T2DM and no other extra condition”; however, from the variable “sum of comorbidities” in Table 2 it is reflected that nearly 5% do have at least one comorbidity. Please check and eventually adjust the wording (e.g., “mainly comprised…”).
10) Related to cluster 4 it is described that it “comprised subjects with cardiovascular conditions, mainly hypertension and dyslipidemia, but without obesity”. Since dyslipidemia is not a cardiovascular disease, please consider using another descriptive term, e.g., “cardiometabolic” instead of “cardiovascular” conditions.
11) There are single typing errors in the text (e.g. subject's instead of subjects' characteristics at p.3; missing “of” in the phrase “by the use electronic health records” at p. 8), please check carefully.
Reviewer 2 Report
The authors reported the results of an observational study that aimed to examine the patterns of morbidities among patients with T2DM. By analyzing the electronic health records of a hospital in Peru, the authors showed that in 9832 individuals with T2DM, obesity, hypertension, and dyslipidemia were common comorbidities, and there existed four clusters of morbidities.
There are some comments.
Comments:
1. Methods (Line 80 on page 2): “We included only Peruvian subjects with diagnosis of T2DM--.” Was the diagnosis of T2DM defined based on the ICD-10 code in a visit or two (or more) visits? Please clarify in more detail.
2. Methods (Line 90 on page 2): “Definitions were based on ICD-10 codes as well as anthropometry measurements and laboratory results-.” Was the diagnosis of morbidities defined based on the ICD-10 code in a visit or two (or more) visits? Were those visits the visits when patients were diagnosed with T2DM? Were the visits when anthropometry or laboratory measurements had conducted the visits when patients were diagnosed with T2DM? Please clarify in more detail.
3. Results (E-Table 2): Were there differences in the prevalence of morbidities comparing those with one, two, or three chronic conditions? A statistical test is suggested here.
4. Discussion: In addition to selection bias, detection bias may also be present in the current study. For instance, T2DM patients with more (or more severe) morbidities may receive more regular hospital visits and thorough examinations and have more morbidities diagnosed than patients with less (or less severe) morbidities. A discussion of this issue is suggested.
